# Quality of Life in Electrochemotherapy for Cutaneous and Mucosal Head and Neck Tumors

**DOI:** 10.3390/jcm10194366

**Published:** 2021-09-24

**Authors:** Giuseppe Riva, Laura Salonia, Elisabetta Fassone, Silvia Sapino, Fabrizio Piano, Giancarlo Pecorari

**Affiliations:** 1Division of Otorhinolaryngology, Department of Surgical Sciences, University of Turin, 10126 Turin, Italy; elisabetta.fassone@unito.it (E.F.); silvia.sapino@unito.it (S.S.); fabrizio.piano@unito.it (F.P.); giancarlo.pecorari@unito.it (G.P.); 2Division of Otorhinolaryngology, Santa Croce Hospital, 10024 Moncalieri, Italy; salonialaura@gmail.com

**Keywords:** electrochemotherapy, head and neck cancer, quality of life, skin cancer

## Abstract

Background: Primary or recurrent head and neck cancer of skin or mucosa represents a challenge for clinicians and could be debilitating for the patient. Electrochemotherapy (ECT) emerged as a local ablative procedure for cutaneous and mucosal head and neck tumors. The aim of this observational study was the evaluation of quality of life (QoL) after ECT in patients without other surgical or radiation options as curative treatment. Materials and methods: The procedure was performed according the ESOPE (European Standard Operating procedure of Electrochemotherapy) protocol. Twenty-seven patients were evaluated before ECT (T0) and 1 (T1), 3 (T2), and 6 (T3) months after the procedure. QoL was assessed by means of the EORTC QLQ-C30 and EORTC QLQ-H&N35 questionnaires. Results: The objective tumor response rate was 48% (11% CR, 37% PR). Bleeding control was achieved in 7/7 patients who experienced bleeding prior to ECT. QoL improvement was observed after the procedure. In particular, global health status and social functioning were higher after ECT (*p* 0.026 and 0.043), while pain, pain-killers use and appetite loss decreased (*p* 0.045, 0.025 and 0.002). Conclusion: ECT represents a safe and effective treatment for skin and mucosal head and neck tumors without other curative options. It ensures a good pain and bleeding control without worsening of QoL.

## 1. Introduction

Primary or recurrent locally advanced head and neck cancer of skin or mucosa represents a challenge for clinicians and could be debilitating for the patient. Surgery and chemoradiation therapy are the main options as curative treatments in such cases. However, in a number of cases they are not feasible because of tumor site or extent and of comorbidities or previous radiotherapy [1].

Electrochemotherapy (ECT) emerged as a local ablative procedure using electroporation for enhanced drug (bleomycin or cisplatin) delivery to tumor cells by generating transient permeation structures in the cell membrane [2]. In the last three decades, ECT showed its effectiveness in the treatment of cutaneous, subcutaneous, mucosal, or deep seated cancer with different histologies [3,4,5,6]. The European Standard Operating procedure of Electrochemotherapy (ESOPE) was published in 2006 and updated in 2018 [2]. ECT has been used with curative or palliative intent for head and neck cancer [3]. It is effective in pain and bleeding control of tumor masses. Advantages includes local cancer control with minimal damage to surrounding healthy tissues, ensuring only a few side effects [4,5]. The objective response rate for cutaneous and mucosal head and neck tumors is 70–80% [4,5].

Quality of life (QoL) is an important treatment in locally advanced head and neck cancer, especially when a palliative intent is attempted [7]. Systemic chemotherapy and/or immunotherapy are the main treatments available for patients without surgical or radiation options. However, they are burdened by severe side effects that may be responsible for treatment breaks or suspension and QoL worsening [8,9].

The primary aim of this retrospective observational study was the evaluation of QoL after ECT in patients with cutaneous or mucosal head and neck cancer who did not have any other surgical or radiation options as curative treatment. Pain and bleeding control was assessed. The results confirmed our hypothesis of an improvement of patients’ QoL with good pain and bleeding control.

## 2. Materials and Methods

Thirty-three patients with cutaneous or mucosal locally advanced head and neck cancer underwent ECT at our department between 2018 and 2020. All the patients did not have any other surgical and/or radiation option with curative intent because of tumor extent or comorbidities. Exclusion criteria were kidney failure, arrhythmia, interstitial lung fibrosis, epilepsy, active infections, a known allergy to bleomycin, previous treatment with bleomycin at the maximum cumulative dosage, and different anticancer therapies administered within 2 weeks of ECT. Ethical review and approval were waived for this study, due to its retrospective nature. Informed consent for the procedure was obtained by all the patients.

The technical procedure was performed according to the ESOPE protocol and its update [2]. The procedures were performed under sedation and local anesthesia. Bleomycin was administered intravenously (15,000 IU/m^2^) before the application of electrical pulses to the target lesions, that started 8 min after the end of bleomycin bolus. Electric pulses were applied by needle electrodes with linear configuration or finger electrodes (IGEA S.p.A., Carpi, Italy), depending on the localization of the cancer. In particular, finger electrodes were used for less accessible mucosal lesions of oropharynx, retromolar trigone or cheek. Electric voltage was delivered with Cliniporator™ (IGEA S.p.A., Carpi, Italy) with the following parameters: 8 pulses of 400 V and 910–1000 V/cm, of 100 μs duration, at 5000 Hz repetition frequency. Multiple insertions (median 18, range 4–32) of the electrode in the tumor tissue were performed to cover the entire lesion and a margin area of free tissue growths of 5 mm around the lesion itself. Treatment was completed within 30 min after first electrode insertion, allowing the maximum concentration of bleomycin within the lesion.

Treatment response was assessed one month after ECT with clinical examination and two months after the procedure with computed tomography for evaluation of deep lesions (oral cavity, oropharynx, and neck nodes). RECIST 1.1 (Response Evaluation Criteria in Solid Tumors) criteria were used [10].

Karnofsky Performance status was used to assess patients’ functional impairments and medical care requirements.

The primary endpoint of our study was the evaluation of quality of life through the European Organization for Research and Treatment of Cancer Quality-of-Life-Questionnaire-C30 (EORTC QLQ-C30) and the European Organization for Research and Treatment of Cancer Quality-of-Life Questionnaire-Head and Neck 35 (EORTC QLQ-H&N35) [11]. The EORTC QLQ-C30 questionnaire consists of 30 items and assesses QoL in cancer patients. It includes a global score, five functional scales, and six symptoms scales. Higher scores for the global health status—QoL and the functional scales indicate a better level of functioning, while higher scores for symptoms scales indicate severe symptoms. The EORTC QLQ-H&N35 is a specific questionnaire for patients with head and neck cancer. It evaluates the severity of symptoms and consists of 35 items. It is divided into 17 symptoms scales, with a higher score indicating more severe symptoms. The “pain killers” item refers to any pain medication.

The subjects were evaluated before ECT (T0) and 1 (T1), 3 (T2), and 6 (T3) months after the procedure. Twenty-seven out of 33 patients were included in the study having at least one post-operative evaluation (T1).

All statistical analyses were carried out using the Statistical Package for Social Sciences, version 20.0. The Kolmogorov–Smirnov test demonstrated a non-Gaussian distribution of variables, so non-parametric tests were used. A descriptive analysis of all data was performed, and they were reported as medians and interquartile range (IQR). The Friedman test was used to assess differences among more than two paired groups in the mean of continuous variables. Post hoc analysis with the Wilcoxon signed-rank test was performed. A *p* < 0.05 was considered statistically significant.

## 3. Results

Median age was 78 years (IQR 15 years). Table 1 reports patients and tumors characteristics. Maximum tumor diameter was greater than 3 cm in 17 patients (63%). Twelve patients with relapse had previous radiation therapy. Reasons for the absence of curative options were: comorbidities (cardiac, pulmonary, hepatic and/or renal), and previous radiotherapy.

Histology, stage and site of electrochemotherapy are highlighted in Table 2. ECT was administered also to metastatic neck adenopathies with skin involvement if present. Needle electrodes with linear configuration were used in 24 cases and finger electrodes in three cases.

Twenty-seven patients had one-month follow-up (T1), whereas only 18 and 11 subjects had 3- (T2) and 6-month (T3) follow-ups. Other patients were lost at follow-up (one patient), underwent other therapies with palliative intent (systemic chemotherapy or immunotherapy for distant progression) after the procedure (three patients), or died before T2 or T3 (12 patients: nine because of tumor progression and three for other causes). Three patients showed a complete response (CR, 11.1%) and 10 cases a partial response (PR, 37.0%) at one-month follow-up, while eight subjects had stable disease (SD, 29.6%) and six cases a progression of the disease (PD, 22.2%), according to RECIST criteria. Figure 1 shows two examples of PR in a patient with T2N0M0 squamous cell carcinoma of preauricular skin and another patient with rT3N0 squamous cell carcinoma of the tongue. These patients did not underwent surgery for comorbidities.

No severe side effects related to ECT were observed. Slight edema in the site of electrode insertion occurred in all the patients and disappeared one week after the procedure. Bleeding control was achieved at T1 in all seven patients who experienced it before ECT.

Table 3 reports Karnofsky Performance status, NRS for pain, EORTC QLQ-C30 and H&N35 questionnaires outcomes (median and interquartile range). Appendix A reports means and standard deviations. Karnofsky Performance status was stable across evaluations (*p* > 0.05). Statistically significant differences over time were observed for EORCT QLQ-C30 Global health status, social functioning, pain, and appetite loss (*p* < 0.05). Post-hoc tests showed significant differences between T0 and T3 for such variables (*p* < 0.05). In particular, an improvement of quality of life was seen (Figure 2) with a decrease of pain and appetite loss. Moreover, H&N35 questionnaire showed a reduced use of pain killers at T3 (*p* < 0.05).

## 4. Discussion

Electrochemotherapy consists of a combination of electroporation and chemotherapy. Brief consecutive electric pulses delivered through electroporation to the tumor cells determines a temporary permeability of the cell membranes, allowing molecules normally impermeable to the cell to diffuse from the extracellular to the intracellular space [12]. Bleomycin is a cytostatic hydrophilic molecule and is the preferred drug for ECT [12]. Indeed, its cytostatic effect is increased by 3- to 700-fold with electroporation [13].

In the last decade, ECT has successfully emerged as a treatment option in cutaneous and mucosal head and neck cancer [2,14,15]. The European Research on Electrochemotherapy in Head and Neck Cancer (EURECA) multi-institutional project reported good outcomes both for skin and mucosal tumors [4,5]. These studies included patients with head and neck cancer whose standard treatments had either failed or were not deemed suitable or declined. Patients with small tumors (<3 cm) were included. The objective response rate was 74–97% for skin cancer (97% for basal cell carcinoma and 74% for other histologies) and 56% for mucosal tumors. Both studies focused on oncologic outcomes, but were reported stable or slightly improved QoL [4,5]. Based on these promising results, further studies have been planned (e.g., ECT as a first line treatment in recurrent squamous cell carcinoma of the oral cavity and oropharynx PDL-1 negative and/or with evident contraindication to immunotherapy [16]).

A recent review about ECT in mucosal head and neck cancer showed that it was used with a palliative intent in 78% of cases with an overall objective response rate of 73% [3]. Longo et al. observed an objective response in 45% of subjects (5% CR, 40% PR) with advanced skin and mucosal head and neck tumors [17]. On the other hand, Pichi et al. showed a 100% overall response (8% CR, 92% PR) in patients with recurrent large lesions of skin or mucosa [18]. Besides oncologic outcomes, QoL represents an important endpoint when treating locally advanced cancer without curative options. Palliative ECT may be performed both in cutaneous and mucosal head and neck tumors. However, only one study evaluated QoL with validated methods in such patients [19].

Plaschke et al. performed a phase II clinical trial in patients with recurrent mucosal head and neck carcinoma without curative treatment options. Secondary endpoints included QoL questionnaires [19]. Twenty-six patients were evaluated showing an objective response rate of 58% (19% CR, 39% PR). EORTC QLQ-C30 and QLQ-H&N35 questionnaires showed a substantial stability of QoL among pre- and post-operative (4 and 8 weeks after the procedure) evaluations without worsening of pain. However, during the necrotic phase (week 2–6), analgesia had to be increased, especially in patients with pre-treatment pain [19].

The EURECA studies on skin and mucosal head and neck cancer analysed QoL by means of EORTC QLQ-C30 and QLQ-H&N35 questionnaires at baseline and after 1, 2, 4, 8, and 12 months. However, they included also small tumors treated with a curative intent [4,5]. Concerning skin cancer, the EURECA project highlighted a significant improvement of physical functioning, role functioning and decrease of fatigue and pain reported by the EORTC QLQ-C30 questionnaire. Moreover, an improvement in all domains of the EORTC QLQ-H&N35 questionnaire was observed, with perception of feeling ill, pain and use of analgesics, and mouth opening being the most significant [5]. Concerning mucosal head and neck tumors, the EURECA projects showed more stable QoL parameters. In particular, the EORTC QLQ-C30 questionnaire was unchanged in all variables, except for diarrhoea. Similarly, the EORTC QLQ-H&N35 questionnaire showed all unchanged scores, except for swallowing with a significantly poorer outcome at 2-month post-operative evaluation. However, patients staying in protocol reported improvement at 4 months regarding swallowing [4].

Although our study included only patients without other curative treatment options (surgery or chemoradiation), the objective response rate was 48% (11% CR, 37% PR), suggesting a crucial role for ECT as a treatment option in these patients. Moreover, no severe adverse effects were recorded. This should be taken into account when choosing the best therapy for patients with locally advanced head and neck tumors and without surgical and/or radiation options. Indeed, systemic chemotherapy is burdened by a high rate of moderate-severe side effects. These impact negatively on patients’ QoL, worsening the psychological burden of a potentially incurable disease [8].

Our results showed a good pain and bleeding control after ECT in locally advanced cutaneous and mucosal head and neck cancer. Furthermore, similarly to the study by Bertino et al. [5], QoL improved after the procedure with long-lasting positive effects on pain control, global health status and social functioning. All the other parameters did not show any worsening after the procedure. Globally, our study demonstrated that QoL improved or remained stable after ECT. The absence of QoL worsening is crucial for patients with advanced cancer, especially for social contact.

The main limits of our study are the heterogeneity of our sample and the lacking of statistical analyses in skin and mucosal cancer subgroups because of small sample size. Heterogeneity includes anatomical location, histology, and TNM status of the patient cohort. Moreover, only 11 patients had a 6-month follow-up. Further studies with larger samples are necessary to better analyse QoL in different subgroups (e.g., skin vs. mucosal cancer, primary vs. recurrent tumors). Moreover, a control arm with the current standard of care will allow a better definition of the treatment modality. The strength is the QoL evaluation at multiple post-operative examinations, demonstrating the long-lasting effect of ECT on QoL, also in difficult cases, such as locally advanced tumors.

## 5. Conclusions

ECT represents a safe and effective treatment for locally advanced skin and mucosal head and neck tumors. It ensures a good pain and bleeding control without worsening of QoL. Global health status improved on average after the procedure both in skin and mucosal cancer. Future studies are mandatory to better assess ECT role as palliative or curative treatment in otherwise untreatable advanced head and neck lesions. Moreover, the combination of ECT and other therapies, such as immunotherapy, should be investigated.

## Figures and Tables

**Figure 1 jcm-10-04366-f001:**
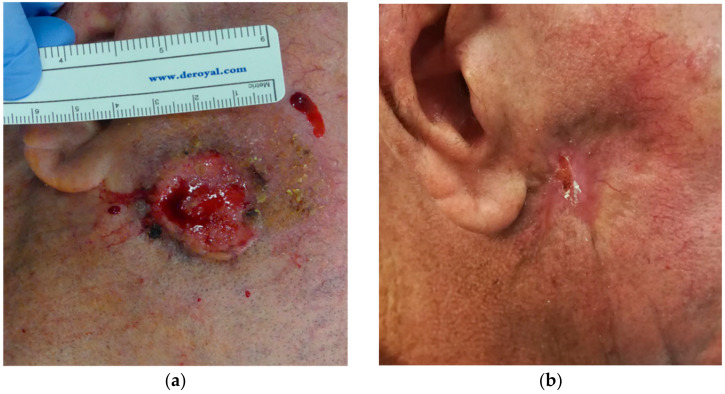
Pre- and post-operative lesions in two patients: (**a**) T2N0M0 squamous cell carcinoma of preauricular skin at T0; (**b**) partial response of preauricolar skin cancer at T3 with bleeding control; (**c**) rT3N0 squamous cell carcinoma of the tongue at T0; (**d**) partial response of tongue cancer at T3.

**Figure 2 jcm-10-04366-f002:**
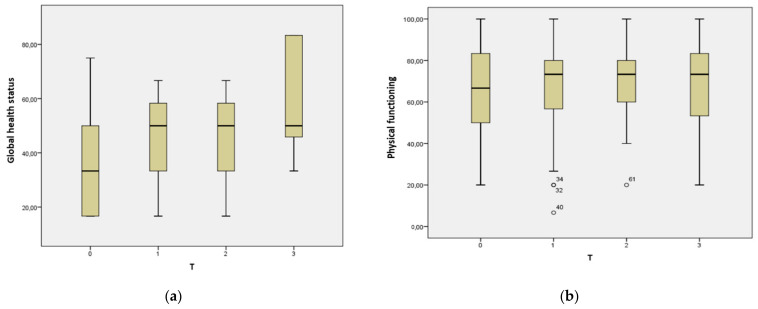
EORCT QLQ-C30 functional scales: (**a**) Global health status; (**b**) Physical functioning; (**c**) Role functioning; (**d**) Emotional functioning; (**e**) Cognitive functioning; (**f**) Social functioning. Post-hoc tests showed significant differences between T0 and T3 for global health status and social functioning (*p* < 0.05). ° and * indicate outliers.

**Table 1 jcm-10-04366-t001:** Patients and tumors characteristics (27 subjects).

Characteristics	N (%)
Sex	
Male	21 (77.8)
Female	6 (22.2)
Smoking	
Active	8 (29.6)
Former	12 (44.4)
Never	7 (25.9)
Alcohol consumption	2 (7.4)
Tumor site	
Skin	9 (33.3)
Mucosa	18 (66.7)
Tumor type	
Primary	11 (40.7)
Relapse	16 (59.3)

**Table 2 jcm-10-04366-t002:** Histology, TNM stage and site of electrochemotherapy (ECT) of the whole cohort (27 patients). Tumor size for basal cell carcinomas was between 3 and 5 cm.

Patients	Site of Primary Tumor	Histology	TNM	ECT Site
1	Skin	Squamous cell carcinoma	T3N3bM0	Inferior lip + adenopathies
2	Skin	Basal cell carcinoma	-	Scalp
3	Skin	Basal cell carcinoma	-	Nose
4	Skin	Basal cell carcinoma	-	Nose
5	Skin	Squamous cell carcinoma	T2N0M0	Preauricolar region
6	Skin	Squamous cell carcinoma	T3N0M0	Preauricolar region
7	Skin	Squamous cell carcinoma	T3N0M0	Preauricolar region
8	Skin	Squamous cell carcinoma	T3N0M0	Preauricolar region
9	Skin	Basal cell carcinoma	-	Cheek
10	Oral cavity	Squamous cell carcinoma	T2N0M0	Retromolar trigone
11	Oral cavity	Squamous cell carcinoma	T2N0M0	Tongue
12	Oral cavity	Squamous cell carcinoma	T3N0M0	Tongue
13	Oral cavity	Squamous cell carcinoma	T0N3bM0	Adenopathies
14	Oropharynx	Squamous cell carcinoma	T2N0M0	Soft palate
15	Oral cavity	Squamous cell carcinoma	T3N0M0	Tongue
16	Oral cavity	Squamous cell carcinoma	T2N0M0	Cheek mucosa
17	Oral cavity	Squamous cell carcinoma	T0N3bM0	Adenopathies
18	Parotid gland	Adenocarcinoma	T4aN0M0	Parotid gland + skin
19	Parotid gland	Adenocarcinoma	T4aN0M0	Parotid gland + skin
20	Oral cavity	Adenocarcinoma	T4aN2aM0	Cheek mucosa + adenopathies
21	Parotid gland	Squamous cell carcinoma	T4aN0M0	Parotid gland + skin
22	Parotid gland	Adenocarcinoma	T4bN3bM0	Parotid gland + adenopathies
23	Parotid gland	Adenocarcinoma	T4bN3bM0	Parotid gland + adenopathies
24	Larynx	Squamous cell carcinoma	T0N3bM0	Adenopathies
25	Oral cavity	Squamous cell carcinoma	T0N3bM0	Adenopathies
26	Oral cavity	Squamous cell carcinoma	T0N3bM0	Adenopathies
27	Oral cavity	Squamous cell carcinoma	T2N0M0	Oral floor

**Table 3 jcm-10-04366-t003:** Karnofsky Performance status, Numeric Rating Scale (NRS) for pain, EORTC QLQ-C30 and H&N35 questionnaires results (median and interquartile range).

Scores	T0	T1 (1 Month)	T2 (3 Months)	T3 (6 Months)	*p* Value
Karnofsky performance status	80.0 (20.0)	80.0 (20.0)	90.0 (30.0)	80.0 (40.0)	0.539
EORTC QLQ-C30 questionnaire					
Global health status	33.3 (33.3)	50.0 (25.0)	50.0 (25.0)	50.0 (41.6)	0.026 *
Physical functioning	66.7 (40.0)	73.3 (26.7)	73.3 (21.7)	73.3 (40.0)	0.596
Role functioning	66.7 (50.0)	66.7 (16.7)	66.7 (16.7)	66.7 (50.0)	0.356
Emotional functioning	75.0 (33.3)	75.0 (16.7)	75.0 (16.7)	33.3 (33.3)	0.243
Cognitive functioning	66.7 (50.0)	33.3 (33.3)	33.3 (33.3)	83.3 (16.7)	0.297
Social functioning	66.7 (50.0)	66.7 (33.3)	66.7 (33.3)	83.3 (16.7)	0.043 *
Fatigue	44.4 (55.6)	33.3 (44.5)	33.3 (25.0)	22.2 (55.6)	0.768
Nausea and vomiting	0.0 (16.7)	0.0 (16.7)	0.0 (16.7)	0.0 (0.0)	0.589
Pain	16.7 (66.7)	16.7 (33.3)	16.7 (33.3)	16.7 (50.0)	0.045 *
Dyspnea	33.3 (33.3)	33.3 (33.3)	33.3 (33.3)	0.0 (33.3)	0.533
Insomnia	33.3 (66.7)	33.3 (33.3)	16.7 (33.3)	0.0 (33.3)	0.138
Appetite loss	33.3 (66.7)	0.0 (33.3)	0.0 (33.3)	0.0 (33.3)	0.002 *
Constipation	0.0 (33.3)	0.0 (33.3)	0.0 (33.3)	0.0 (33.3)	0.179
Diarrhea	0.0 (33.3)	0.0 (0.0)	0.0 (0.0)	0.0 (33.3)	0.072
Financial difficulties	0.0 (0.0)	0.0 (0.0)	0.0 (0.0)	0.0 (33.3)	0.697
H&N35 questionnaire					
Pain	16.7 (16.7)	16.7 (25.0)	16.7 (27.1)	0.0 (16.7)	0.049 *
Swallowing	8.3 (25.0)	8.3 (41.7)	8.3 (35.4)	0.0 (25.0)	0.642
Senses problems	16.7 (33.3)	16.7 (33.3)	16.7 (33.3)	0.0 (16.7)	0.436
Speech problems	22.2 (33.3)	22.2 (33.3)	22.2 (36.1)	22.2 (22.2)	0.742
Trouble with social eating	8.3 (25.0)	16.7 (33.3)	16.7 (33.3)	16.7 (25.0)	0.869
Trouble with social contact	6.7 (20.0)	13.3 (33.3)	10.0 (33.3)	6.7 (20.0)	0.452
Less sexuality	0.0 (66.7)	33.3 (66.7)	33.3 (66.7)	33.3 (66.7)	0.392
Teeth	0.0 (33.3)	0.0 (33.3)	0.0 (33.3)	0.0 (33.3)	0.585
Opening mouth	33.3 (66.7)	33.3 (33.3)	16.7 (41.7)	33.3 (33.3)	0.632
Dry mouth	0.0 (33.3)	0.0 (33.3)	0.0 (0.0)	33.3 (33.3)	0.145
Sticky saliva	33.3 (33.3)	33.3 (33.3)	16.7 (33.3)	33.3 (33.3)	0.768
Coughing	0.0 (33.3)	0.0 (0.0)	0.0 (0.0)	0.0 (0.0)	0.190
Felt ill	0.0 (0.0)	0.0 (33.3)	0.0 (33.3)	0.0 (0.0)	0.787
Pain killers	100.0 (100.0)	100.0 (100.0)	100.0 (100.0)	0.0 (100.0)	0.025 *
Nutritional supplements	0.0 (100.0)	0.0 (100.0)	0.0 (100.0)	0.0 (100.0)	0.066
Feeding tube	0.0 (0.0)	0.0 (0.0)	0.0 (0.0)	0.0 (0.0)	0.392
Weight loss	0.0 (100.0)	0.0 (100.0)	0.0 (100.0)	0.0 (100.0)	0.234
Weight gain	0.0 (0.0)	0.0 (0.0)	0.0 (0.0)	0.0 (0.0)	0.330

* *p* < 0.05.

## Data Availability

The data presented in this study are available on request from the corresponding author.

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
