# Peer review of "Quality of Life in Electrochemotherapy for Cutaneous and Mucosal Head and Neck Tumors"

_jcm, 2021, doi:10.3390/jcm10194366_

Round 1

Reviewer 1 Report

I commend the authors on conducting an interesting study assessing quality of life outcomes in head and neck cancer patients undergoing ECT. This is an especially relevant topic given that ECT is generally reserved for patients with advanced cancer lacking other treatment options, where quality-of-life is one of the primary goals. I think that this study could be suitable for publication if the following concerns are addressed:

  1. Abstract: Please change “objective response rate” to “objective tumor response rate” to clarify that it refers to tumor response, not the questionnaire response rate. I recommend removing “pain and bleeding control was good” from the abstract as this statement is very vague. Instead, you could include something along the lines of “bleeding control was achieved in 7/7 patients who experienced bleeding prior to ECT.” Finally, I would recommend including p-values in parentheses next to the quality-of-life outcomes.

  1. Results, page 4, line 125: Can the authors please list the specific number of patients who were (1) lost to follow-up, (2) undergoing other palliative therapies, and (3) died before T2 or T3.

  1. Table 3: Can the authors please include the number of follow-up months in parenthesis next to T1, T2, and T3. For example, T2 (3 months). Otherwise, readers will have to keep referring to the methods section to clarify.

  1. Table 3: Do the authors have any insight into why the cognitive functioning scores dropped so low for T1 and T2, and then came back up for T3?

  1. Although it is a non-parametric analysis, reporting the median and IQR in Table 3 makes it difficult to interpret changes in scores across timepoints because the sample size is so small. I would recommend that the authors provide a similar table with mean and standard deviation at each timepoint. This could either be provided as a supplemental table or as a separate row below each variable in the current table.

  1. The Numeric Rating Scale (NRS) for pain does not add anything to the manuscript since the two quality of life questionnaires already have pain scales. Furthermore, the non-significant p-value for the NRS outcome contradicts the findings for pain in the QoL scales. I would recommend removing this outcome measure from the manuscript.

  1. For “pain killers” in the H&N35 questionnaire, can you clarify in the methods section about how this information was elicited in the survey (e.g. does it refer to opioids, NSAIDs, acetaminophen, any pain medication in general, etc.)?

  1. There is potential for significant selection bias given the loss to follow-up (only 11/27 or ~41% of patients responding at T3). For example, if patients lost to follow-up were in an overall poorer state of health, this could explain the apparent improvement on the QoL scales. I would recommend performing a sensitivity analysis for the significant variables in Table 3 only including the 11 patients who completed full follow-up. This should be included in the results section and addressed as one of the limitations in the discussion section.

Reviewer 2 Report

This  study “ Quality of life in electrochemotherapy for locally advanced cutaneous and mucosal head and neck tumors”  is  interesting, however there are many bias related to the methods and results.

The study selected patients with locally advanced skin and mucosal head and neck tumor without other surgical or radiation options as curative treatment. However they  included in the study patients with staging T2 N0 M0, that are not locally advanced. Can the authors explain why they were included these patients and why  they did’nt treated them with surgery or surgery plus chemo/radiotherapy? Have these patients previously treated and with which kind of treatement? The reason for palliative treatment in these T2N0M0 tumors has not been specified. What were the contraindications to surgical or radiation therapy in these patients, and the comorbidities?

Why were the T0N3bM0 patients  not able to have a rescue neck dissection? Can the authors explain?

Lines 84-86 How  the follow-up  was done after the first two months? Have they make CT, physical examination? Can the authors clarify?

Lines 124-127  Clarify why patients at 3 months were 18 and at six months they were 11. The 7 patients died, were they treated differently? How many in patients lost to follow up have died. Why did they die?

How many patients lost in follow-up have changed palliative treatment and why? Can the authors clarify?

Figure 1

Because the lesions shown in the photographs were not treated surgically?

Table 2

Neither TNM nor tumor size is reported in some skin cancers. Can authors add at least the dimention?.

Table 3

How many patients does the table refer to? Only the eleven patients evaluated after 1 month, 3 months, 6 months should be considered.

The assessment of pain relief in such a small sample of patients (3 patients) cannot be taken into account to draw valid conclusions.

How was the bleeding assessment performed? How many bleeding lesions were there before the treatment?

Lines 164  For bibliographic completeness authors should add this document: Domanico R, Trapasso S, Santoro M, Pingitore D, Allegra E. Electrochemotherapy in combination with chemoradiotherapy in the treatment of oral carcinomas in advanced stages of disease: efficacy, safety, and clinical outcomes in a small number of selected cases. Drug Des Devel Ther. 2015 Feb 19;9:1185-91.

Round 2

Reviewer 1 Report

The authors have adequately addressed all of the previous comments.

Reviewer 2 Report

After clarifications and corrections in the text, the manuscript can be accepted.

This manuscript is a resubmission of an earlier submission. The following is a list of the peer review reports and author responses from that submission.